# “It All Starts by Listening:” Medical Racism in Black Birthing Narratives and Community-Identified Suggestions for Building Trust in Healthcare

**DOI:** 10.3390/ijerph22081203

**Published:** 2025-07-31

**Authors:** Jasmine Y. Zapata, Laura E. T. Swan, Morgan S. White, Baillie Frizell-Thomas, Obiageli Oniah

**Affiliations:** 1Department of Pediatrics, University of Wisconsin-Madison, Madison, WI 53792, USA; jzapata@uwhealth.org; 2Reproductive Equity Action Lab, Department of Population Health Sciences, University of Wisconsin-Madison, Madison, WI 53726, USA; 3Department of Family Medicine and Community Health, University of Wisconsin-Madison, Madison, WI 53705, USA; morgan.white@fammed.wisc.edu; 4Department of Family Medicine, University of Colorado, Aurora, CO 80045, USA; baillie.frizell@cuanschutz.edu; 5School of Medicine and Public Health, University of Wisconsin-Madison, Madison, WI 53792, USA; oniah@wisc.edu

**Keywords:** medical distrust, provider bias, racial discrimination

## Abstract

This study documents Black Wisconsinites’ birthing experiences and their proposed solutions to improve Black birthing people’s trust in healthcare. Between 2019 and 2022, we conducted semi-structured, longitudinal interviews (both individual and focus group interviews) with those enrolled in a local perinatal support group program for Black birthing people (*N* = 25), asking about their pregnancy, birthing, and postpartum experiences and their ideas for building trust in healthcare. Using the Daughtering Method and Braun and Clarke’s method of reflexive thematic analysis, we coded the interview data and then iteratively collated the codes into themes and subthemes. Participants described experiencing medical racism, including healthcare trauma and provider bias, during pregnancy and delivery. They drew connections between those experiences and the distrust they felt toward healthcare providers and the healthcare system. They provided actionable strategies that individual providers and the healthcare system can take to build the trust of Black birthing people: employ more Black providers, listen to Black birthing people, exhibit cultural humility, engage in shared decision-making, build personal connections with patients, and spend more time with patients. This study connects Black birthing people’s experiences of medical racism to feelings of medical distrust and provides community-identified actionable suggestions to build trust and shape how we combat racial disparities in healthcare provision and health outcomes.

## 1. Introduction

Racial disparities in pregnancy-related and infant health outcomes are a profound public health issue. In the United States, Black birthing people are 3 to 4 times more likely to die from pregnancy-related causes than their white counterparts [1]. Black birthing people are often preoccupied not only with their own health but also that of their children, given that the U.S. Black infant mortality rate is 2.4 times higher than that of white infants [2]. These disparities are, in part, due to inequities created by structural racism, which is built into systems from housing and education to criminal justice and healthcare [3,4].

One way that racism in perinatal healthcare manifests is interpersonally, through discriminatory care (i.e., unfair or unequal treatment of patients based on their race/ethnicity [5]). Specifically, one study estimated that one in eight non-Hispanic Black respondents experienced racial discrimination during their perinatal care [6]. In another study, over one in five non-Hispanic Black respondents reported experiencing racial discrimination during childbirth [7]. Additionally, a body of qualitative work has documented ways in which anti-Black racism manifests in healthcare settings through stereotyping, invalidation, poor communication, neglectful care, dismissal by healthcare professionals, and an absence of shared decision-making [8,9,10,11,12,13,14,15].

These race-related health disparities and discriminatory healthcare experiences are compounded by centuries of unethical and exploitative treatment in healthcare and beyond, leading to a profound sense of medical distrust for many Black Americans [16]. This distrust in the healthcare system and in healthcare providers, and the resulting hesitancy to engage with medical care, is well-documented [8,17,18,19,20,21,22,23]. For example, in one study, Black women’s perceptions of racial discrimination in medical encounters, and even their anticipation of discrimination, led them to minimize the frequency of their healthcare interactions [17].

Given the historical context in which Black Americans seek perinatal care, the ways that discriminatory care has eroded trust in the healthcare system, and the links between this trust and health-seeking behaviors, the healthcare system must invest in building trust among Black Americans. There is a need to generate innovative solutions to heal relationships between the healthcare system and the patients that they have alienated through generations of systemic and interpersonal wrongdoings. It is especially appropriate to seek such solutions from the very communities that have been most impacted by medical racism and therefore have critical insights into the steps required to begin to heal these relationships.

Drawing from the stories of Black birthing people in Wisconsin, where Black infant mortality rates are among the highest in the nation [24], this study documents the lived experiences of Black birthing people and their proposed solutions to improve the relationships between Black birthing people and the healthcare system.

## 2. Materials and Methods

### 2.1. Today Not Tomorrow Pregnancy and Infant Support Program

In close consultation with community partners, the first author developed and facilitated the Today Not Tomorrow Pregnancy and Infant Support Program (TNT-PISP). The name is inspired by one of the group’s community partners, Today Not Tomorrow, Inc., which aims to support families in utilizing their own strengths and proactively creating healthy families [25]. The TNT-PISP combined community-based doula programs, group-based models of prenatal care, and community-based pregnancy support groups to provide perinatal care and support for Black birthing people and infants in Dane County, Wisconsin. Located in the south–central part of Wisconsin, Dane County is an urban area that includes the state capital, Madison [26]. As of 2024, its population was estimated at 588,347, with a racial and ethnic composition of about 84% White, 6% Black or African American, 7% Asian, and 8% Hispanic or Latino [27].

We identified and recruited TNT-PISP participants through local community partners that serve pregnant people and Black families in Dane County. A total of 25 birthing people enrolled and participated in the TNT-PISP for at least one month during their prenatal and postpartum periods. The group met monthly from October 2019 to August 2022. It was initially an in-person support program, but it transitioned to a virtual group in early 2020 due to the onset of COVID-19. Additional details about the TNT-PISP are being published elsewhere.

### 2.2. Data Collection

We collected data using both individual and focus group interviews, conducting semi-structured, longitudinal interviews with the 25 Black birthing people who participated in the TNT-PISP.

We aimed to complete three individual interviews for each participant throughout their prenatal and postpartum period, with the first interview taking place early in the pregnancy, the second interview in their early postpartum period, and the third interview when they were around 6 months postpartum. A total of 11 participants completed individual interviews between October 2019 and March 2021, with some participants (*n* = 6) completing only one interview and others completing two (*n* = 3) or three interviews (*n* = 2). We followed an interview guide to ask participants about their pregnancy, birthing, and postpartum experiences and their ideas for building Black birthing people’s trust in the healthcare system. Most interviews lasted approximately 90 minutes, although some took as little as 40 minutes or as much as 3 hours.

We also conducted three 90-minute virtual focus groups. A total of 14 TNT-PISP attendees participated in one of the first two focus groups. The third focus group was held with people who had already participated in another interview, either individual or focus group. The focus groups covered the same material as the individual interviews. We held the first focus group in the same month that the TNT-PISP began (October 2019), the second one a month later (November 2019), and the final one after the conclusion of the program (August 2022).

We compensated participants with $20 for each individual or focus group interview that they participated in. The first author, a Black mother and a physician in Dane County, facilitated all individual and focus group interviews. With participants’ permission, we audio-recorded all interviews, which were later transcribed verbatim by the research team and a professional transcription service. The Institutional Review Board at the University of Wisconsin-Madison approved these study procedures.

### 2.3. Data Analysis

We grounded our data collection and data analysis in Evans-Winters’ Daughtering Method [28]. The Daughtering Method is a Black feminist methodological framework designed to decolonize qualitative research and center the lived experiences, wisdom, and resilience of Black women. It emphasizes relational accountability, self-reflection, and the honoring of ancestral knowledge, encouraging researchers to approach inquiry not only as a technical process but as a deeply ethical and spiritual practice.

Our research team studied the Daughtering Method and collaboratively developed a protocol to guide our work. This protocol involved intentional practices such as reflection, movement, nourishment, mindfulness, and cleaning the air with positive energy before engaging with participants or data. These practices were designed to ground researchers in care, clarity, and cultural humility. We also built our interview guide to center strength and resilience, ensuring that our questions did not pathologize participants’ experiences but instead invited narratives of agency, insight, and transformation. This approach shaped both how we engaged with participants and how we interpreted their stories, reinforcing our commitment to non-extractive, community-rooted research.

Our data analysis also followed Braun and Clarke’s method of reflexive thematic analysis [29,30]. We trained a medical student and a family medicine resident in the Daughtering Method and in Braun and Clarke’s analytic method. They independently double-coded each transcript under the supervision of the second author, a PhD-educated researcher trained in qualitative methods. After familiarizing themselves with the data, they systematically generated initial codes across the entire data set, identifying and labeling chunks of data that represented key ideas related to our research questions. Then, the research team met to collate codes into themes and subthemes, reconciling codes and refining the themes and subthemes iteratively until we reached consensus, had defined and labeled the themes and subthemes, and had generated a thematic map highlighting the overall story produced through the qualitative inquiry. In addition to these reconciliation meetings, the research team met biweekly throughout the data analysis process to discuss coding progress and challenges.

## 3. Results

As described below, our analysis (1) reveals patterns in Black birthing experiences related to experiencing medical racism and (2) outlines Black birthing people’s suggestions for how to build trust in the healthcare system.

### 3.1. Medical Racism in Black Birthing Experiences

Many of our participants described experiencing medical racism, including healthcare trauma and provider bias, during their perinatal care. Participants detailed a variety of distressing or disturbing experiences while interfacing with the healthcare system. Often, these experiences were related to a lack of appropriate communication during emergency labor and delivery situations. For example, one participant shared her lack of agency during her labor process as a source of trauma:

“It was just—this one lady took away all of my power, and then when I fought for it afterwards because I was angry and I was scared. And when I fought for it afterwards, I was painted as somebody—even going into it, the day that I went into labor, because I wanted a spontaneous labor, and so the c-section was not urgent, but, like, they wanted to get it done faster. I was painted as a difficult patient.”

Participants commonly labeled such experiences as “traumatic” and spoke about the ways that healthcare trauma caused them to avoid future healthcare seeking and make alternative plans in the future, such as arranging for a home birth.

One participant recounted, “I have heard many stories from women of color who said, like, their birth plan wasn’t really acknowledged or respected by their physician or their health team. They were talked down to or treated like they weren’t smart.”

Another participant described the emotional toll of being dismissed by providers, even in urgent situations, explaining that Black birthing people often have to “say, ‘no, you’re not about to discharge me.’ Like, ‘I’m not OK. Something is not right,’ and having to go to that extreme to be heard…that’s why I say maybe physicians not believing Black women when they say, like, ‘I’m in abnormal pain.’ I don’t know if it’s because people assume that we just do this birthing thing differently than other women or we have a higher tolerance of pain, or they’re just the expert and we’re just the person feeling the pain.”

Many participants shared about times when their provider made assumptions about them or their family based on their race or ethnicity. In one participant’s words, such bias is “not blatant, but it’s like subtle things. And it’s like if you’re not paying attention or if you’re not aware, then you’re going to…miss it. But it’s just little things…but it’s there.”

As described by another participant, “I feel like with providers, sometimes I’ve gone into the doctor’s office and 100% felt like they were making so many assumptions about who I was.”

Another participant reflected on how these assumptions shaped the care they received: “The discrimination and racism you feel even when you do have access and are connected with a provider—your provider thinks of you as, like, a stereotype, talks down to you… those negative interactions can really have an influence on those outcomes and also the labor and delivery.”

Participants believed that healthcare providers should work to address their biases; as one participant suggested, “be aware that those [biases] can show up in their medical practice and kind of take those extra steps to kind of, I don’t know, remove those biases and have a more equitable practice across all of their, all their patients and all their families.”

Participants’ experiences with medical racism affected the way that they felt about and interacted with the healthcare system in the future. Participants explained that these experiences created doubts about the honesty and reliability of providers and the healthcare system. Specifically, one participant explained feeling that “white doctors…really don’t, I don’t feel have my best interest at heart.”

Many participants shared a specific distrust of white providers, and for others, this distrust applied more generally to non-Black providers. One participant shared,

“So I probably wouldn’t trust them [doctors who aren’t Black] at all. There’s probably nothing they can do to make me trust them, to make me more comfortable or nothing like that. I would probably ask them 50 million questions. I would always go back to somebody I know to go on the internet, research, things like that.”

Although some participants expressed a distrust of non-Black providers that would be virtually impossible to overcome, other participants described ways that healthcare providers could build their trust by improving the patient-provider relationship (see below).

### 3.2. Strategies for Building Trust

Participants offered a variety of suggestions for ways that healthcare providers and the healthcare system can build trust among Black birthing people (see Figure 1): (1) employ more Black providers, (2) listen to Black birthing people, (3) exhibit cultural humility, (4) engage in shared decision-making, (5) build personal connections with patients, and (6) spend more time with patients. In the words of one of our participants, “I get that it can be hard sometimes, but you’re also doctors, you learn, you’re constantly learning and you’re constantly keeping up to date with new information…You can adapt.”

#### 3.2.1. Employ More Black Providers

Related to a common distrust of non-Black healthcare providers (discussed above), participants expressed a desire to be seen by Black healthcare providers while also reflecting on the lack of available Black providers locally. One participant sought out a Black provider based on the assumption that a provider with a shared racial identity would understand her needs and experiences, describing,

“And so, to be able to talk to a provider who gets it and like understands that layer of stress that’s constantly on top of everything else, it’s really nice to just be able to talk with someone about that and understanding being a person of color, and so, my primary care doctor, she is a person of color, and that’s why I sought her out, and I have a wonderful—every time I go to her for a visit, it’s so nice, it’s like therapeutic in a way.”

Similarly, another participant shared, “I knew that she got me on an unspoken level that I wouldn’t have if it had been a white provider. And so…yeah, I think having more OB-GYNs [and] midwives of color.” Other participants reiterated these points, stating that, for Black birthing people, racial concordance with their provider can “bring a sense of, like, commonality or connectiveness” and make their conversations feel less “awkward” or “uncomfortable” because of “having to provide a ton of explanation” in order for a white provider to understand their needs and experiences. Participants described feeling that “a Black doctor, they might do their job and even go above and beyond to make sure you and your child is actually ok.”

Despite participants’ perceptions of the importance of patient-provider racial concordance, participants described finding a Black provider in their community as “near impossible.” Participants identified this as a key area for healthcare system improvement. As one participant stated, “We need more people our color in [the field of medicine] so we feel more safe.”

#### 3.2.2. Listen to Black Birthing People

Almost universally, participants expressed that Black birthing people are not listened to in their interactions with healthcare providers. In fact, multiple participants compared conversations with providers to a “lecture” or being “talk[ed] at” with no room for compromise. Specifically, many participants shared stories of their birth plans being ignored or disrespected. For example, one participant explained, “I wasn’t listened to and [the provider] took everything that I had worked for, that I had planned for, away from me in one, in, like, one 20-minute meeting with her.” Others reiterated these points, sharing stories of providers “refusing to let me labor at all” or even “consider it whatsoever.” Despite the potential medical issues involved in these cases, the participants were distressed by their perception of not being heard and “listen[ed to] with an open mind.”

One participant reiterated the importance of listening to Black patients, stating,

“I do get that obviously the doctors are well-trained, well-educated, but I also believe that the patient, the person in front of you, also knows their body better than you do and better than anyone else does. So, if they are coming in with a viewpoint that you should listen, you should definitely listen and either try and compromise or work through it as opposed to just dismissing the cares of that person or actively going against what the client or the patient wants simply because of your own internal biases.”

One participant described her experience, emphasizing her choice to change providers when she did not feel listened to:

“And it doesn’t matter how well you speak to your provider, how eloquently you can say what you need, you’re just a Black body. I mean, I know from myself that I’ve gone in to see providers that I realized that this will be the last time that I see this person, and I will find someone else. They’re not even listening to anything I’m saying or anything that I know that is going on in my body.”

Participants highlighted specific areas where providers need to listen to Black birthing people, including their level of pain and their drive to protect themselves and their children. Often, participants connected their perceptions of being ignored in the healthcare system to the high rates of Black infant and pregnancy-related mortality. For example, one participant explained that she believes the reason for racial disparities in pregnancy-related deaths is because of “physicians not believing Black women when they say, like, ‘I’m in pain’ or ‘this isn’t normal.’” Another participant expanded on this point, stating,

“We have to believe Black moms when we say that we’re, you know, in pain or we have to recognize that little Black boys aren’t just strong and big, but they’re also precious and tender and in need of, you know, that, the same care that someone would give to a little girl or a little child that’s not Black.”

Participants expressed that Black birthing people were the experts of their own bodies, and healthcare providers, particularly white providers, needed to listen to Black birthing people when they shared their concerns, beliefs, and needs. As one participant summarized, healing and building trust with Black patients “all starts by listening.”

#### 3.2.3. Exhibit Cultural Humility

In their interactions with non-Black healthcare providers, participants emphasized the need for providers to exhibit an attitude of openness and understanding of differences in thoughts, attitudes, and actions that stem from different cultural backgrounds. One participant pointed out that “cultural education is extremely important” for healthcare providers. Another reiterated that providers should be prepared “to engage with folks from different cultures and races and ethnicities.”

Participants reflected that cultural humility was needed for providers to better understand Black birthing people as patients, with one participant sharing, “I think I would just need to feel more confident that non-Black people understand that the experiences of Black folks is not the same as theirs.”

Participants also reflected on providers’ need to understand how parenting differs between cultures, saying, “providers can, even if they’re not Black, having some sort of, I don’t know, cultural competence or some ability to understand why people parent the way they do, it’s because of their culture.” One participant shared a specific example in which she was “shamed” by a provider for “bed sharing.” She explained that the conversation was “really intense” and the provider “seemed like she didn’t really see me or the whole me.” Instead, the patient wished that her provider could “at least acknowledge the cultural factor of why I was doing it” or had provided “some suggestions of how to make this bed sharing safer.” Instead, the patient was left “just so turned off” and unable to “trust” the provider.

Participants also pointed out that it is the job of the healthcare system and not individual Black patients to educate providers and ensure that they are prepared to meet their patients’ needs. Participants specifically recommended that the medical “curriculum need[s] to be revamped [to] talk about the sterilization that was done to Black women, talk about the history, the abuse, and let people really sit with that.”

#### 3.2.4. Engage in Shared Decision-Making

Participants also spoke of a need for equal and reciprocal patient-provider relationships in which healthcare plans are made together. For example, one participant summarized what she was looking for throughout her pregnancy, stating,

“And my nurses were really informative. They were giving me the ins and outs of everything because that’s what I needed. I needed to know, ‘what are you doing?,’ ‘Why are you doing it?,’ ‘What is it going to do for me?,’ ‘How is it going to help me?,’ ‘How is it going to affect [my baby] before she’s born?’ And I think having providers be able to do that even more throughout your—the process, especially when you’re going to appointments. And just giving you the details necessary, that you need to feel comfortable and know like, oh, I do have options.”

Participants described the desire for supportive providers who treat patients “with respect and dignity” and help them “make informed decisions” about prenatal care and birth, reiterating that this helps them have “trust and confidence” in their healthcare.

#### 3.2.5. Build Personal Connections with Patients

Participants discussed how important it was to feel that their providers went beyond medical care to ensure that they felt respected and cared for as a person, especially when trying to build trust with non-Black providers. One participant shared,

“I’ve noticed a difference in providers who, like, when I’m starting to work with them, they ask me not just what I do for work, but they ask me about my family and where I’m from and how I’m enjoying or how living in [this city] is. Just having more of a personal connection with me really helps me to build trust with the provider. And I’ve also had providers who, like, really just jumped into the visit and don’t really take the time to engage, ask, like a real person, develop any sort of connection. And so, for those providers, I just am less likely to speak up or to want to talk with them because it seems like they don’t really care. So just like at a basic level, taking time to get to know someone.”

Other participants reiterated these points, with one suggesting concrete strategies for providers looking to build personal connections:

“Like, if it’s good eye contact and they pronounce names correctly, or they’re asking questions like, ‘how are you doing’ and listening, and then maybe the next time they come in the room, they follow up…But I think it just goes back to being seen and being heard, and feel like you see me as an individual, and not just the third delivery of the day.”

#### 3.2.6. Spend More Time with Patients

Participants expressed a desire for more time with providers, referring both to the time they spent with a provider during an appointment and also the need for a more sustained relationship with one provider over the course of their pregnancies. They acknowledged the constraints that healthcare providers are under, with one participant stating, “I know that providers are often limited in their time that they have with each person. And so appointments can’t be very long.” Despite acknowledging these time constraints, participants expressed frustrations at receiving “rushed” care and not having “time to really ask those questions or to ask them for the explanation about what’s going on.”

Several participants shared that they had selected midwives for their primary obstetrical care due to the increased face-to-face time permitted. As one participant shared,

“Our [midwife] appointments were so comprehensive…our appointments would be each like an hour long or more, an hour or more long. And she would ask me about my entire being.”

Regarding the lack of continuity in obstetrical care, one participant noted, “Again, it’s really hard at [health system name redacted] because I feel like I was always seeing different people.” Another participant reiterated this point, stating, “one of the things that I did not like was how, when I was in labor…there’s just so many people coming in and out of the room, and I had developed a relationship with my OB/GYN, but that wasn’t the person who I saw at all during my labor and delivery. It felt like a loss to not be able to have that continuity of who my main person was when the big day came.”

## 4. Discussion

This study draws from the stories of Black birthing people in Wisconsin to highlight the medical racism commonly experienced by Black patients during pregnancy and delivery, draw connections between those experiences and the profound sense of medical distrust felt by many Black Americans, and provide actionable strategies that individual healthcare providers and the healthcare system can take in order to build Black birthing people’s trust in healthcare.

These stories echo other accounts of medical racism during pregnancy and delivery, highlighting common experiences of inadequate or inappropriate provider communication and a lack of respect for Black birthing people’s bodies [8,10,11,12,31]. Our participants astutely connected the racism they experienced during pregnancy and delivery to their sense of healthcare distrust. Others have written extensively on this topic, describing how medical racism erodes patient trust and impacts if and how patients seek healthcare in the future [12,16,17,18,19,20,21,22,32,33].

Much of the existing literature on medical racism calls for interventions or lays out researcher-developed suggestions to improve disparities in perinatal healthcare and reduce anti-Black healthcare discrimination. This study goes a step further by highlighting Black birthing people’s own suggestions for how individual healthcare providers and the healthcare system can build Black Americans’ trust.

First and foremost, participants highlighted the need for Black healthcare providers. As described in our study and in the existing literature [8,32], many Black birthing people prefer to see a Black provider. Often, this was discussed not as a casual preference but as a response to prior experiences of medical racism and as an important act of resistance, self-advocacy, and self-defense. Emerging research supports this inclination, showing that patient-provider racial concordance is associated with improved health outcomes for Black patients [34]. These findings have important system-level implications. Given that less than 6% of US physicians identify as Black or African American [35], medical institutions seeking to build the trust of Black Americans should invest in educating, employing, and retaining providers of color.

Acknowledging the challenges of increasing diversity among healthcare providers, especially in areas that lack overall diversity, such as our study setting in Madison, Wisconsin, our participants provided other suggestions for providers of any race to help build strong relationships with Black birthing people. They described specific provider qualities that made them feel heard, respected, and cared about, explaining the need for shared decision-making and cultural humility from providers. They also highlighted specific ways that spending time with a given provider can build a sense of safety and trust. When describing a desire for more time with providers, participants explained a desire for both an increased amount of time within a single appointment as well as a sustained relationship with the same provider over the course of the pregnancy. Given that providers face increasing administrative burdens and pressures to fit more patients into tighter appointment times, these tangible requests can only be achieved with healthcare system-level buy-in.

### Strengths and Limitations

As we work to combat infant and pregnancy-related health inequities, it is critical that we include and elevate the voices of those most impacted. This study draws from Black birthing people’s lived experiences to seek solutions from one of the very communities most impacted by medical racism. Our participants have highlighted actionable steps for the medical community to implement in order to begin building trust among Black Americans and ultimately shape how we combat racial inequities in infant and reproductive health. Our study also benefited from a longitudinal design, enhancing the depth and reliability of our findings by capturing participants’ evolving experiences, perspectives, and behaviors over time.

These findings should be interpreted in the context of the study’s limitations. Notably, these data were collected from 2019 to 2022, which coincides with the height of the COVID-19 pandemic as well as other national events related to racism in the police system that garnered public attention and response. Obtaining stories of medical racism and healthcare suggestions during this distinctive time in history provides a unique perspective on these issues. However, this context undoubtedly shaped participants’ experiences and perspectives in ways that are not necessarily generalizable to all healthcare contexts. Additionally, the pandemic briefly interrupted our study procedures and caused some study participants to be lost to follow-up. These disruptions limited the amount of data we could draw from and forced us to move to virtual data collection.

## 5. Conclusions

This study connects Black birthing people’s experiences of medical racism to feelings of medical distrust and provides community-identified actionable suggestions to build trust and shape how we combat racial disparities in healthcare provision and health outcomes.

## Figures and Tables

**Figure 1 ijerph-22-01203-f001:**
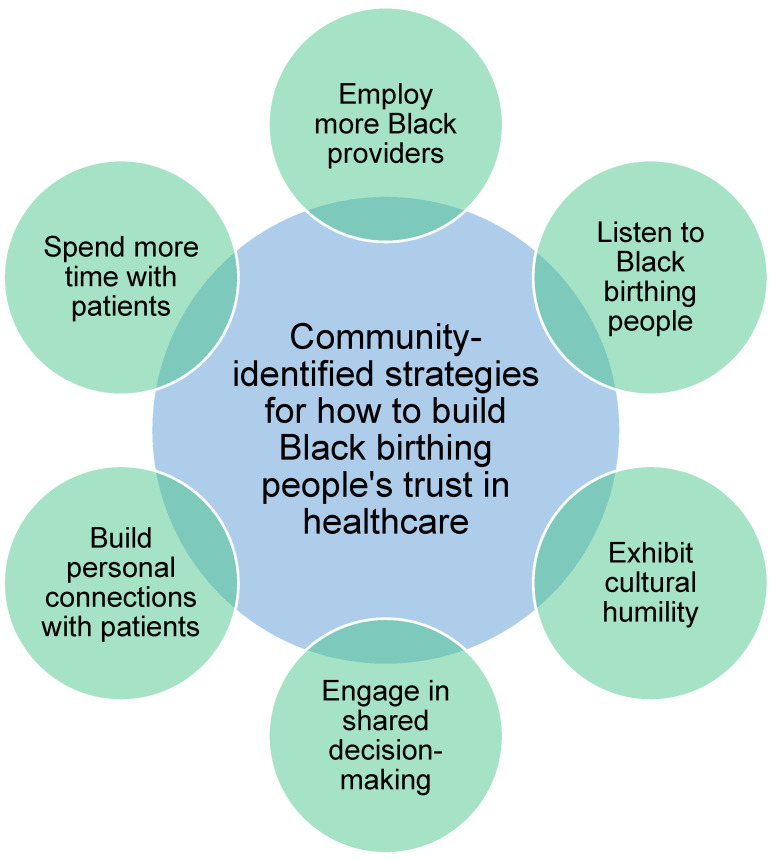
Six strategies for how to build Black birthing people’s trust in healthcare, identified through qualitative interviews with Black birthing people in Wisconsin.

## Data Availability

Data are not publicly available due to privacy and ethical restrictions.

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
