# Peer review of "“It All Starts by Listening:” Medical Racism in Black Birthing Narratives and Community-Identified Suggestions for Building Trust in Healthcare"

_ijerph, 2025, doi:10.3390/ijerph22081203_

Round 1
Reviewer 1 Report
Comments and Suggestions for Authors
This manuscript reports on experiences of racism of Black women during obstetric care, with a focus on proposed solutions. The article is very well written, and the focus on solutions is a new addition to the literature, and I definitely recommend publication. I have only a few suggestions that may help to strengthen what is already a very strong manuscript.
- From the Methods it appears that both interviews and focus groups were conduced and the data used in the analysis. I advise including in the abstract that the design includes both, whereas currently the abstract only mentions interviews.
- The design included longitudinal interviews, and the authors describe well how disruptions and loss-to-follow up prevented a purely longitudinal design. Yet, I am really curious if for those participants that did have longitudinal interviews, anything of interest arose from talking with them over time. Even though it was not all participants, it is a really unique strength of the study.
- Do the authors have sociodemographic characteristics of the participants to describe?
- Related to the comment above, it would be helpful to provide a brief description of Dane County to international readers to understand better the context. Is it a largely urban and surburban county, or did the sample include also rural areas?
Reviewer 2 Report
Comments and Suggestions for Authors
Dear Authors,
Thank you for submitting your manuscript. Below are my suggestions and comments, intended to help improve the clarity, structure, and overall quality of your work."
"I hope you find these remarks constructive and helpful in revising the manuscript."
-
Abstract;
-
Keywords should be listed in alphabetical order.
-
Introduction;
-
“discriminatory care” should be defined.
-
…… medical distrust for many Black Americans (A source or citation should be added.)
-
Respectful Maternity Care could have been mentioned.
-
Materials and Methods;
-
How was the sample size calculated, and is the sample size sufficient?
Data Collection;
-
A brief explanation about the research questions should be provided
-
With participants’ permission (Oral and written consent)
Data Analysis;
-
The Daughtering Method suggests processes to aid in decolonizing qualita- tive methods and centering Black women’s issues (A reference should be included).
-
Discussion;
-
Emerging research supports this inclination, showing that patient-provider racial concordance is associated with improved health outcomes for Black patients [25].
-
It should be supported by the relevant literature. The discussion should be expanded with reference to the literature.
-
The limitations section should be provided
Reviewer 3 Report
Comments and Suggestions for Authors
The authors work from the assumption that racism/discrimination is embedded in the healthcare system and their respondents experience it. There is ample research to support this. Based on this assumption, they ask their respondents to share their experiences as as patients care for by healthcare professionals. However, they actually do not have strong evidence that these birthing people have experienced medical racism (p4). For example, one respondent reported "I have heard many stories..." This is actually not evidence of medical racism -- 1) it was experienced by the informant and 2) is a "story" or "rumor" about something that may or may not be evidence of medical racism. On this same page, another reports "I feel like with providers, sometimes I've gone into the doctor's office and 100% felt like they were making so many assumptions about who I was." This too, is not an account of medical racism because the description of the person's experiences are absent -- what did the healthcare provider say or do, for example, that led the respondent to feel this way? Without this description, the authors are not doing what they set out to do -- that is do provide readers with a "study that documents the lived experiences of Black birthing people and their proposed solutions...." I can say much more here but this type of evidence appears throughout the manuscript and I encourage the authors to re-read each quote they provide to make certain that is an "experience" and not a summary of a feeling or thought.
It may be helpful to know more about the healthcare providers as well -- are respondent's experiences different when providers are not Black? How? Do the authors have recordings of experiences that evidence differences by provider demographics that could be incorporated? If so, is their advice for healthcare providers different by demographics (e.g. race, gender -- which is also determinant of healthcare)
The project is an important one and I urge the authors to further develop this work. From my view, it is about bringing clarity to the research question (introduction) further grounding it in existing literature, and better aligning the methodology (including greater description about "daughtering methods" with the question (at present this is unclear to the reader and in fact no evidence of this is in the evidence that I see).
In addition, and perhaps most significant -- the manuscript will be greatly improved by more descriptive data that directly supports the research question/thesis/purpose (e.g. documentation of the birthing people's "experiences" with medical racism and then their proposed solutions). Right now, this is mostly absent.
Reviewer 4 Report
Comments and Suggestions for Authors
- You have conducted such a critical study, and centering the voices of Black women in both their experiences with racism and suggestions for addressing the problem is needed in the literature.
- Several notable researchers on this topic are not cited, including Joia Crear Perry, Monica McLemore, and Karen Scott. I suggest reviewing their work and adding their essential contributions to the Background section
- There should be increased clarity around your definition of racism. You mention structural racism in the opening of the paper, but much of your evidence of racism in the results section is more consistent with personally-mediated racism. Increased discussion of the terminology will be helpful for readers to contextualize the findings.
- Citation needed on line 44
- Citation needed on line 55
- Citation needed on line 90
